# Polymorphisms in ACE, ACE2, AGTR1 genes and severity of COVID-19 disease

**Maria Sabater Molina**[1,2,3,4]*, **Elisa Nicolás Rocamora**[1], **Asunción Iborra Bendicho**[5], **Elisa García Vázquez**[6], **Esther Zorio**[3,7], **Fernando Domínguez Rodriguez**[3,4,8], **Cristina Gil Ortuño**[1,2], **Ana Isabel Rodríguez**[1], **Antonio J. Sánchez-López**[9], **Rubén Jara Rubio**[10], **Antonio Moreno-Docón**[5], **Pedro J. Marcos**[3,11], **Pablo García Pavía**[3,4,8,12], **Roberto Barriales Villa**[3,13], **Juan R. Gimeno Blanes**[1,2,3,4]

**1** Cardiac Department, Hospital Universitario Virgen de la Arrixaca, Murcia, Spain, **2** Cardiogenetic Laboratory, Instituto Murciano de Investigación Biosanitaria, Murcia, Spain, **3** Centro de Investigación Biomédica en Red Cardiovascular (CIBERCV), Madrid, Spain, **4** European Reference Network for Rare and Low Prevalence Complex Diseases of the Heart (ERN Guard-Heart), Madrid, Spain, **5** Microbiology Department, Hospital Universitario Virgen de la Arrixaca, Murcia, Spain, **6** Department of Infectious Disease, Hospital Universitario Virgen de la Arrixaca, Murcia, Spain, **7** Cardiology Department, Unidad de Cardiopatías Familiares y Muerte Súbita, Hospital Universitario y Politécnico La Fe, Valencia, Spain, **8** Department of Cardiology, Hospital Universitario Puerta de Hierro, Madrid, Spain, **9** Biobanco Hospital Universitario Puerta de Hierro/Instituto de Investigación Sanitaria Puerta de Hierro-Segovia de Arana, Madrid, Spain, **10** IntensiveCareUnit, Hospital Universitario Virgen de la Arrixaca, Murcia, Spain, **11** Dirección Asistencial, Servicio de Neumología, Complejo Hospitalario Universitario de A Coruña (CHUAC), Instituto de Investigación Biomédica de A Coruña (INIBIC), A Coruña, Spain, **12** Universidad Francisco de Vitoria (UFV), Pozuelo de Alarcón, Madrid, Spain, **13** Cardiac Department, Complejo Hospitalario Universitario de A Coruña, A Coruña, Spain

* mariasm79es@hotmail.com

**Data Availability Statement:** All relevant data are within the manuscript and its Supporting Information files.

## Abstract

### Background

Infection by the SARS-Cov-2 virus produces in humans a disease of highly variable and unpredictable severity. The presence of frequent genetic single nucleotide polymorphisms (SNPs) in the population might lead to a greater susceptibility to infection or an exaggerated inflammatory response. SARS-CoV-2 requires the presence of the ACE2 protein to enter in the cell and ACE2 is a regulator of the renin-angiotensin system. Accordingly, we studied the associations between 8 SNPs from AGTR1, ACE2 and ACE genes and the severity of the disease produced by the SARS-Cov-2 virus.

### Methods

318 (aged 59.6±17.3 years, males 62.6%) COVID-19 patients were grouped based on the severity of symptoms: Outpatients (n = 104, 32.7%), hospitalized on the wards (n = 73, 23.0%), Intensive Care Unit (ICU) (n = 84, 26.4%) and deceased (n = 57, 17.9%). Comorbidity data (diabetes, hypertension, obesity, lung disease and cancer) were collected for adjustment. Genotype distribution of 8 selected SNPs among the severity groups was analyzed.

### Results

Four SNPs in ACE2 were associated with the severity of disease. While rs2074192 andrs1978124showed a protector effectassuming an overdominant model of inheritance

**Funding:** The work was in part supported by grants from Instituto de Salud Carlos III and FEDER Union Europea, Una forma de hacer Europa (FONDO-COVID19 COV20/00420). The group of researchers is part of the Cardiovascular Research Network (CIBERCV) and the CIBER of Rare Diseases (CIBERER). The research group in "Hereditary Heart Diseases and Sudden Death" is registered at the University of Murcia and the IMIB. The Family Heart Disease Unit of the Virgen de la Arrixaca University Clinical Hospital is accredited as a Reference Unit (CSUR) by the Ministry of Health and is part of the network of European reference centers included in Guard-Heart (ERN). María Sabater has a research contract from the FFIS (Foundation for Sanitary Training and Research).

**Competing interests:** The authors have declared that no competing interests exist.

(G/A vs. GG-AA, OR = 0.32, 95%CI = 0.12–0.82; p = 0.016 and A/G vs. AA-GG, OR = 0.37, 95%CI: 0.14–0.96; p = 0.038, respectively); the SNPs rs2106809 and rs2285666were associated with an increased risk of being hospitalized and a severity course of the disease with recessive models of inheritance (C/C vs. T/C-T/T, OR = 11.41, 95% CI: 1.12–115.91; p = 0.012) and (A/A vs. GG-G/A, OR = 12.61, 95% CI: 1.26–125.87; p = 0.0081). As expected, an older age (OR = 1.47), male gender (OR = 1.98) and comorbidities (OR = 2.52) increased the risk of being admitted to ICU or death vs more benign outpatient course. Multivariable analysis demonstrated the role of the certain genotypes (ACE2) with the severity of COVID-19 (OR: 0.31, OR 0.37 for rs2074192 and rs1978124, and OR = 2.67, OR = 2.70 for rs2106809 and rs2285666, respectively). Hardy-Weinberg equilibrium in hospitalized group for I/D SNP in ACE was not showed (p<0.05), which might be due to the association with the disease. No association between COVID-19 disease and the different AGTR1 SNPs was evidenced on multivariable, nevertheless the A/A genotype for rs5183 showed an higher hospitalization risk in patients with comorbidities.

## Conclusions

Different genetic variants in ACE2 were associated with a severe clinical course and death groups of patients with COVID-19. ACE2 common SNPs in the population might modulate severity of COVID-19 infection independently of other known markers like gender, age and comorbidities.

## Introduction

The SARS-CoV-2 virus causes a severe and fatal infection in certain patients, mostly, but not exclusively, in elderly individuals with significant preexisting conditions. Hypertension is one of the most consistent predictors of mortality [1, 2]. The fact that SARS-CoV-2 requires the presence of the ACE2 protein to enter in the cell membrane [3] and the association between a higher rate of complications in hypertensive patients, have suggested that angiotensin converting enzyme inhibitors or angiotensin II receptor blockers could facilitate the first phase of viral infection. But perhaps these same drugs could be beneficial in the inflammatory phase of the disease. Chronic therapy with angiotensin system agentsis known to cause changes in the expression of ACE, ACE2, and AGTR1 [3–5].

Some authors, have suggested that blocking ACE2 as a potential strategy to reduce viral SARS-CoV-2 load in the pneumocytes, preventing spreading into other organs [6]. On the contrary, inhibition of ACE2 in already infected COVID-19 patients, could be deleterious via the consequent decrease in the production of angiotensin 1–7, which has been demonstrated to play an anti-inflammatory and antifibrotic activity through its receptor (MasR) [7–9].

Infection by the SARS-Cov-2 virus produces in humans a highly variable disease of unpredictable severity. Some individuals are completely asymptomatic while others end up, after a chain series of infection and inflammatory processes, going through distress, microvascular-thrombosis, multi-organ failure to death [10]. Despite the identified prognostic factors, there is great unexplained variability [11].

It is possible that the differences in the activity of certain proteins, conditioned by the presence of frequent polymorphic genetic variants in the population, might lead to a greater susceptibility to infection, a greater efficiency of viral replication, or to an exaggerated

inflammatory response [12]. It is reasonable to speculate that the presence of several polymorphisms of the ACE (I/D), ACE2 (rs2074192, rs1978124, rs2074809, rs2074666) and AGTR1 (rs5183, rs5185, rs5186) genes could explain both, the propensity to infection, the extension to different organs and the degree of the severity of the COVID-19 clinical presentations [13, 14].

Accordingly, we aimed to study the associations between eight AGTR1, ACE2 and ACE gene polymorphisms and the severity of the disease produced by the SARS-Cov-2 virus.

## Methods

### Research ethics considerations

This study was conducted in accordance with the principles of the 1975 Declaration of Helsinki and approved by the Ethics and Scientific Committees of each participating institutions [Hospital UniversitarioVirgen de la Arrixaca and, BIOBANC-MUR (Murcia), Biobank Hospital Universitario y Politécnico la Fe (Valencia), Biobank Hospital Universitario de A Coruña (A Coruña), Biobank Hospital Universitario Puerta de Hierro Majadahonda (Madrid), Biobank Hospital Clínico San Carlos (Madrid)]. Informed written consent was obtained from all patientsor their relatives.

### Study subjects

A total of 318COVID-19 subjects with positive polymerase chain reaction (PCR) test for SARS-Cov-2 virus were included in the study. The kit used for PCR test was Novel Coronavirus (2019-nCoV) Real Time Multiplex RT-PCR kit (Detection for 3 Genes), manufactured by Shanghai ZJ Bio-Tech Co., Ltd. (Liferiver) and the CFX96 Touch Real-Time PCR Detection System (BioRad). The participants were grouped into 4 groups: outpatients cured, hospitalized on the wards, admitted to the Intensive Care Unit (ICU) and deceased as a result of the infection or its complications. Patients were selected consecutively from those with available samples from the 5 participating centers' biobanks, with the aim to achieve a minimum of 50 cases per group.

To carry out the study of polymorphisms and haplotypes, DNA was extracted from 400 μl of peripheral blood samples using the Maxwell® 16 Blood DNA Purification Kit (Promega). The study of the I/D polymorphism in ACE was carried out by PCR protocol using an initial denaturation at 94˚C for 5 min; 30 cycles of denaturation at 94˚C for 1 min, annealing at 64˚C for 45 sec, and elongation at 72˚C for 1 min. The final cycle was followed by extension at 72˚C for 5 min and electrophoresis in agarose (2%) gel. The remaining selected single nucleotide polimorphisms (SNPs) in ACE2 and AGTR1 genes were analyzed by Sanger sequencing procedures. PCR reactions were performed in a final volume of 25 μL containing 2 μL of DNA and using a touchdown PCR protocol: initial denaturation at 94˚C for 5 min, followed by 10 touchdown cycles (0.2˚C decrease of annealing temperature every cycle) and 35 standard cycles: denaturation for 1 min at 94˚C, primer annealing for 35 sec at 62˚C, and primer extension for 30 sec at 72˚C. The last cycle was followed by 5 min incubation at primer extension temperature of 72˚C. After they were purified and sequenced on a DNA 3500XL Genetic Analyzer (Applied Biosystems).

### SNPs in genes of renin-angiotensin system included in the study

Different SNPs were selected in base to previous studies where they were related with mortality in acute respiratory distress syndrome or pneumonia as the case of the I/D polymorphism in ACE [15, 16]. Several SNPs in ACE2 have been investigated as risk factors for hypertension and heart failure, such as rs2106809, an important predictive factor of the response to

**Table 1. Different SNPs of renin-angiotensin system included in this study.**

| Chr | Gen | Locus | rs | Variant | Type | MAF (gnomAD) |
|-----|-----|-------|-----|---------|------|--------------|
| X | ACE2 | NG_012575 | rs2074192 | g.42403G>A | Intron | 0.42428 |
| | | | rs1978124 | g.7130A>T | Intron | 0.37498 |
| | | | rs2106809 | g.7132T>C | Intron | 0.19141 |
| | | | rs2285666 | g.14845G>A | Intron | 0.280049 |
| 3 | AGTR1 | NG_008468 | rs5183 | NG_008468.1:g.49227A>G NM_000685.4:c.1062A>G | Synonymous Variant | 0.061514 |
| | | | rs5185 | NG_008468.1:g.49315T>G | 3' UTR | 0.026048 |
| | | | | NM_000685.4:c.*70 = | | |
| | | | rs5186 | NG_008468.1:g.49331A>C | 3' UTR | 0.227483 |
| | | | | NM_000685.4:c.*86 = | | |
| 17 | ACE | NG_011648 | rs4646994[a] | Intrón 16 | Intron | [b] II 48.1% |
| | | | I/D 287 pb | | | ID 40.5% |
| | | | | | | DD 11.5% |

Chr: Chromosome.

[a]rs number for this polimorphism was not found in dbSNP and therefore no reported allele frequencies were available for comparison.

[b]Frequencies were obtained from Küçükarabaci B, 2008 [23] and Bellone M, 2020 [24]. MAF: Minor allele frequency.

antihypertensive treatment with ACE inhibitors [17] or rs2074192 and rs2106809 were also associated with risk for left ventricular hypertrophy [18]. In addition, rs2285666 SNP, has been recently related to a lower COVID-19 infection as well as case-fatality rate among Indian populations [19]. The rs5186 (C) allele in AGTR1 is associated with increased risk for essential hypertension [20, 21]. Age and gender may also influence risk of AGTR1 SNPs and their role in hypertension and related disorders [22].

The **Table 1** shows the frequencies of the different SNPs from renin-angiotensin system (RAS) included in this study.

## Statistical and bioinformatics analysis

Hardy-Weinberg equilibrium (HWE) was assessed by the $\chi2$ test. I/D polymorphism in ACE failed HWE and it was removed from analysis. The level of significance adopted was $p>0.05$ in outpatients group. For SNPs from ACE2 females and males were analyzed separately since the ACE2 gene is located on the X chromosome. Allele frequencies were calculated according to the genotypes of all patients.

Using Pearson chi-square test or Fisher's exact probability (for categorical variables), the variations in frequency distribution of genotypes and demographic characteristics (gender, age and comorbidities) were assessed. The association strength was calculated applying odds ratios (ORs) and 95% confidence intervals (CIs). All genetic models were evaluated, including dominant, recessive, co-dominant, overdominant, and log additive models of inheritance for seven SNPs with SNPStats software (https://www.snpstats.net/start.htm). Each model provides different assumptions regarding the genetic effect. Using the SNPStats, haplotype frequencies were also obtained for ACE2 and AGTR1 according to the expectation maximization algorithm [25]. Statistical analyses were done by SPSS version 23.0 (IBM, Chicago, USA). The forest-plot was performed by multivariable analysis including age, gender and comorbidities with each one of the SNPs.

The statistical tests were assumed significant for $p<0.05$. The potential impact of each SNP on the functional protein was analyzed using the in silico software Mutation Taster [26].

## Results

### General characteristics of the study subjects

318 patients with documented COVID-19 positive PCR were classified based on the severity of symptoms: Outpatients (n = 104. 32.7%); hospitalized on the wards (n = 73, 23.0%), admitted to ICU (n = 84, 26.4%) and deceased (n = 57, 17.9%). Sex, age and comorbidities data are presented in **Table 2**.

Mean age was significantly different between groups, being higher in the deceased. The gender was significantly different between groups being the proportion of males higher than females in the ICU but not in the deceased. Regarding comorbidities, the proportion of hypertension, diabetes and obesity showed significant differences between groups. The percentage of patients with hypertension was lower in outpatients compared with the rest of groups, particularly with deceased group (20.4% vs 49.1%). Diabetes was more prevalent within ICU and deceased patients. Percentage of obesity was higher in hospitalized on the wards group (26.4%).

In order to analyze genotype distributions of different SNPs, all patients were grouped in two: outpatients (n = 104) and hospitalized patients (n = 214, which included hospitalized on the wards, ICU and deceased patients) (**Table 2**).

The p values for the HWE are showed in **S1 Table**. It should be taken into account that the HWE might not be met in the case sample, which could be indicative that the SNP may be associated with the disease, such in case of I/D polymorphism in ACE.

### Prediction of functional impacts of SNPs on protein function and stability

The rs1978124 and rs2106809 are located at the beginning of the intron 2; rs2285666 is located in intron 3 and rs2074192 in the intron 16. Analysis of these four SNPs in ACE2 gene by in silico software Mutation Taster [26] to explore their impacts on splicing process, and consequently ACE2 function have demonstrated that rs2074192 leads to increased donor site and protein features might be affected. Correspondingly, rs1978124 and rs2285666 lead to creation of new donor splice site. No potential effect on splicing was showed by rs2106809.

**Table 2. Characteristics of each group of patients included in the study.**

| | | Total (n = 318) | Outpatients (n = 104) | On the wards (n = 73) | Intensive Care Unit (n = 84) | Deceased (n = 57) | P value | Hospitalized[b] (n = 214) | P value | ICU + deceased (n = 141) | P value |
|---|---|---|---|---|---|---|---|---|---|---|---|
| Age (mean ±SD) | | 59.6±17.3 | 52.7±17.5 | 60.1±16.4 | 57.6±13.5 | 74.6±14.0 | <0.0001 | 63.1±16.2 | <0.0001 | 64.6±16.0 | <0.0001 |
| Gender[a] | Male | 198 (62.3%) | 59 (56.7%) | 43 (58.9%) | 65 (77.4%) | 31 (54.4%) | 0.007 | 139 (65.0%) | 0.175 | 96 (68.6%) | 0.080 |
| | Female | 119 (37.4%) | 45 (43.3%) | 30 (41.1%) | 18 (21.4%) | 26 (45.6%) | | 74 (34.6%) | | 44 (31.4%) | |
| Comorbidities | | 196 (63.6%) | 48 (50.5%) | 49 (67.1%) | 52 (62.7%) | 47 (82.5%) | 0.001 | 148 (69.5%) | 0.002 | 99 (70.7%) | 0.002 |
| Hypertension | | 105 (34.1%) | 19 (20.4%) | 31 (43.1%) | 27 (32.5%) | 28 (49.1%) | 0.001 | 86 (40.4%) | <0.0001 | 55 (39.3%) | 0.003 |
| Diabetes | | 44 (14.3%) | 4 (4.3%) | 11 (15.3%) | 17 (20.5%) | 12 (21.1%) | 0.005 | 40 (18.8%) | <0.0001 | 29 (20.7%) | <0.0001 |
| Obesity | | 40 (13.0%) | 5 (5.4%) | 19 (26.4%) | 8 (9.6%) | 8 (14.0%) | 0.001 | 35 (16.4%) | 0.006 | 16 (11.4%) | 0.161 |
| Chronic lung disease | | 27 (8.8%) | 5 (5.4%) | 10 (13.9%) | 9 (10.8%) | 3 (5.3%) | 0.172 | 22 (10.3%) | 0.191 | 12 (8.6%) | 0.444 |
| Cancer | | 29 (9.4%) | 7 (7.5%) | 8 (11.1%) | 5 (6.0%) | 9 (15.8%) | 0.212 | 22 (10.4%) | 0.528 | 14 (10.0%) | 0.642 |

[a] n = 1 missing gender.

[b] Hospitalized = hospitalized on the wards + ICU + deceased patients. ICU: intensive care unit.

Rs5183, rs5185 and rs5186 SNPs also might affect the AGTR1 function. Rs5185 and rs5186 SNPs are located in the region 3'UTR and result in two and one, respectively, new donor splice sites, and increased acceptor and donor splice sites. Rs5183 is located at the end of codifying region with no aminoacid change and leads to increased acceptor splice site.

## The association between ACE2 and AGTR1 SNPs and hospitalization risk

Three SNPs of ACE2 rs2074192, rs2106809 and rs2285666 were associated with hospitalization in females (**Table 3** and **S2 Table**); while rs2106809 and rs2285666 were associated with an increased risk of being hospitalized in the log-additive and recessive models of inheritance [(OR = 2.12, 95% CI: 1.00–4.52; p = 0.039) and (OR = 6.56, 95% CI: 0.71–60.20; p = 0.048)], respectively, it was found that rs2074192 showed a protector effect, assuming an overdominant model of inheritance. Rs2074192 G/A genotype was significantly associated with a lower risk of hospitalization (G/A vs. GG-AA, OR = 0.40, 95%CI: 0.17–0.92; p = 0.029).

No association between COVID-19 patent setting (outpatient vs hospitalized) disease and the different AGTR1 SNPs was demonstrated.

## The association between ACE2 and AGTR1 SNPs and severity of the disease (outpatiens vs ICU+deceased)

Considering that the reason for hospitalization on the words in some patients might not be due only to the severity of the COVID-19 disease, as rather to the expected "a priori" risk of

**Table 3. Genotype and allele frequencies of ACE2 and AGTR1 SNPs in hospitalized and non-hospitalized COVID-19 cases.**

| Locus | Model | Genotype | Outpatients (n = 104) | Hospitalized (n = 214) | Odds Ratio | p-value |
|---|---|---|---|---|---|---|
| **ACE2 FEMALE** (n = 119, adjusted by age + comorbidities) | | | | | | |
| **rs2074192** | Overdominant | G/G-A/A | 14 (31.1%) | 41 (56.2%) | 1.00 | **0.029** |
| | | G/A | 31 (68.9%) | 32 (43.8%) | 0.40 (0.17–0.92) | |
| **rs1978124** | Overdominant | A/A-G/G | 22 (48.9%) | 45 (61.6%) | 1.00 | 0.13 |
| | | A/G | 23 (51.1%) | 28 (38.4%) | 0.53 (0.23–1.21) | |
| **rs2106809** | Log-additive | | | | 2.12 (1.00–4.52) | **0.039** |
| **rs2285666** | Recessive | G/G-G/A | 43 (97.7%) | 66 (89.2%) | 1.00 | **0.048** |
| | | A/A | 1 (2.3%) | 8 (10.8%) | 6.56 (0.71–60.20) | |
| **ACE2 MALE** (n = 190, adjusted by age + comorbidities) | | | | | | |
| **rs2074192** | --- | G/G | 33 (64.7%) | 87 (62.6%) | 1.00 | 0.86 |
| | | A/A | 18 (35.3%) | 52 (37.4%) | 1.06 (0.52–2.17) | |
| **rs1978124** | --- | G/G | 26 (51%) | 74 (53.2%) | 1.00 | 0.68 |
| | | A/A | 25 (49%) | 65 (46.8%) | 0.87 (0.44–1.71) | |
| **rs2106809** | --- | T/T | 43 (84.3%) | 103 (74.1%) | 1.00 | 0.24 |
| | | C/C | 8 (15.7%) | 36 (25.9%) | 1.68 (0.70–4.05) | |
| **rs2285666** | --- | G/G | 44 (86.3%) | 108 (77.7%) | 1.00 | 0.29 |
| | | A/A | 7 (13.7%) | 31 (22.3%) | 1.62 (0.64–4.10) | |
| **AGTR1** (n = 309, adjusted by age + gender + comorbidities) | | | | | | |
| **rs5183** | Recessive | A/A-A/G | 96 (100%) | 212 (99.5%) | 1.00 | 0.21 |
| | | G/G | 0 (0%) | 1 (0.5%) | NA (0.00-NA) | |
| **rs5185** | --- | T/T | 95 (99.0%) | 210 (98.6%) | 1.00 | 0.7 |
| | | T/G | 1 (1%) | 3 (1.4%) | 1.59 (0.14–17.53) | |
| **rs5186** | Log-additive | | | | 0.72 (0.48–1.08) | 0.12 |

Genotype- and allele type-specific risks obtained in the best model of inheritance based in Akaike information criterion (AIC). OR, odds ratio; CI, confidence interval; SNPs, single nucleotide polymorphisms.

complications based on medical history and comorbidities, or availability of hospital beds, etc, in this section of the paper we present genotype distributions of different SNPs in the "outpatients" (n = 104) versus "ICU + deceased" groups (n = 141). **Table 2** shows the clinical characteristics of these two groups of patients. Age and proportion of patients with comorbidities were significantly higher in the ICU + deceased combined group, being hypertension and diabetes the main comorbidities associated with the risk of admission to the ICU or deceased.

There was an association of the different ACE2 SNPs with the severity of the disease when the female group of outpatients and the female group of greater severity (ICU + deceased) were compared (**Fig 1**, **Table 4** and **S3 Table**). Genotype G/A of rs2074192 and genotype A/G of the rs1978124 SNPs in the overdominant model were overrepresented in the outpatient group with an OR = 0.32 (0.12–0.82), p = 0.016 and OR = 0.37 (0.14–0.96), p = 0.038, respectively. On the contrary, the risk of being admitted in the ICU or dying was higher in COVID-19 patients who presented the C/C for the rs2106809 SNP (OR = 11.41, 95%IC: 1.12–115.91, p = 0.012) and A/A genotype for rs2285666 SNP (OR = 12.61, 95%IC: 1.26–125.87, p = 0.0081), both in the recessive models. The distribution of different genotypes of the AGTR1 SNPs did not show significant differences between the disease severity groups.

### Interaction analysis with comorbidities

As expected, comorbidities increased the risk of belonging to the hospitalization group; nevertheless, the interaction analysis showed that particular genotypes modulated the magnitude of this association. Some genotypes in selected SNPs were overrepresented in the more severe group while others were more prevalent in the milder outpatient group (**S4 Table**).

The interaction analysis of comorbidities and genotypes of SNPs in ACE2, showed the G/A genotype from rs2074192 in females conferred a protective factor for those patients without comorbidities (OR = 0.13, 95% CI: 0.03–0.52, p: 0.019) (**S4 Table**). On the other hand, those females with G/A genotype for rs2074192 and comorbidities had higher risk of hospitalization (OR = 1.27, 95%CI: 0.22–7.19, p<0.0001).

These interactions between comorbidities and the genotype in rs2074192 of ACE2 were more evident when female outpatients vs ICU + deceased groups were compared (p = 0.0074) (**S5 Table**).

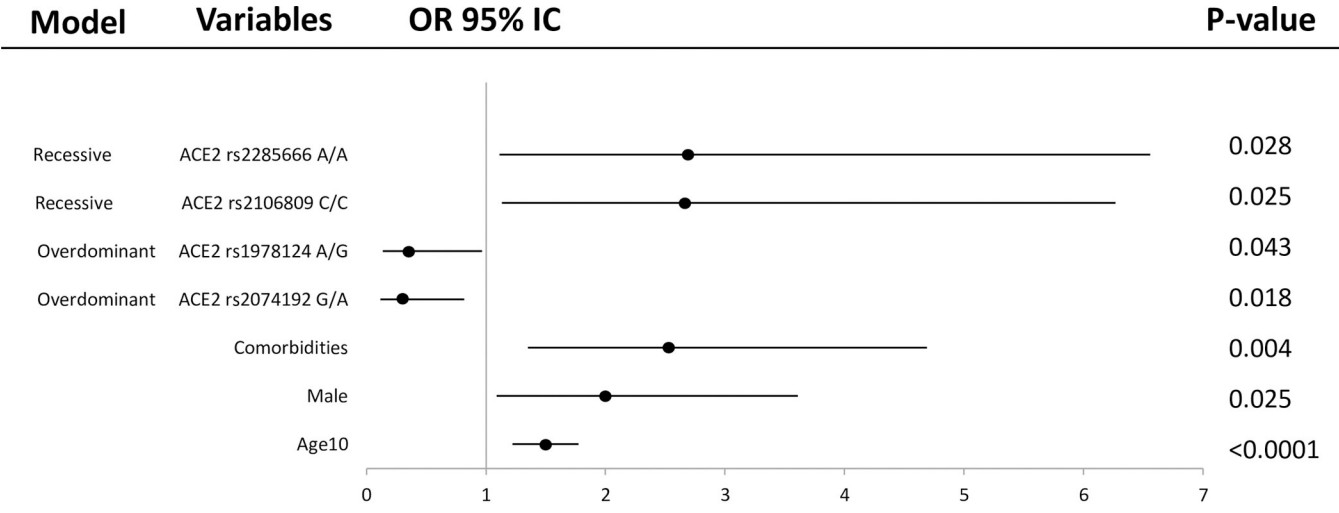

**Fig 1. Forest plot of different covariates included in the association study of selected SNPs for estimation of OR of ICU+deceased vs outpatient individuals.** The horizontal lines correspond to the study specific OR and 95% CI. Each of the SNPs were included separately in different models with age(10), gender and comorbidities as covariates. Age(10) represents OR per 10 years increase.

**Table 4. Genotype and allele frequencies of ACE2 and AGTR1 SNPs in outpatients and ICU+deceased COVID-19 cases.**

| Locus | Model | Genotype | Outpatients (n = 104) | ICU+deceased (n = 141) | OR (95% CI) | P-value |
|---|---|---|---|---|---|---|
| **ACE2 FEMALE** (n = 88, adjusted by age + comorbidities) | | | | | | |
| **rs2074192** | Overdominant | G/G-A/A | 14 (31.8%) | 26 (59.1%) | 1.00 | **0.016** |
| | | G/A | 30 (68.2%) | 18 (40.9%) | 0.32 (0.12–0.82) | |
| **rs1978124** | Overdominant | A/A-G/G | 20 (46.5%) | 30 (68.2%) | 1.00 | **0.038** |
| | | A/G | 23 (53.5%) | 14 (31.8%) | 0.37 (0.14–0.96) | |
| **rs2106809** | Recessive | T/T-T/C | 42 (97.7%) | 37 (84.1%) | 1.00 | **0.012** |
| | | C/C | 1 (2.3%) | 7 (15.9%) | 11.41 (1.12–115.91) | |
| **rs2285666** | Recessive | G/G-G/A | 43 (97.7%) | 37 (84.1%) | 1.00 | **0.0081** |
| | | A/A | 1 (2.3%) | 7 (15.9%) | 12.61 (1.26–125.87) | |
| **ACE2 MALE** (n = 147, adjusted by age + comorbidities) | | | | | | |
| **rs2074192** | --- | G/G | 33 (64.7%) | 63 (65.6%) | 1.00 | 0.91 |
| | | A/A | 18 (35.3%) | 33 (34.4%) | 0.96 (0.43–2.10) | |
| **rs1978124** | --- | A/A | 26 (51%) | 50 (52.1%) | 1.00 | 0.68 |
| | | G/G | 25 (49%) | 46 (47.9%) | 0.86 (0.41–1.81) | |
| **rs2106809** | --- | T/T | 43 (84.3%) | 70 (72.9%) | 1.00 | 0.16 |
| | | C/C | 8 (15.7%) | 26 (27.1%) | 1.93 (0.75–4.96) | |
| **rs2285666** | --- | G/G | 44 (86.3%) | 73 (76%) | 1.00 | 0.22 |
| | | A/A | 7 (13.7%) | 23 (24%) | 1.82 (0.68–4.86) | |
| **AGTR1** (n = 235, adjusted by age + gender + comorbidities) | | | | | | |
| **rs5183** | --- | A/A | 87 (90.6%) | 122 (87.8%) | 1.00 | 0.67 |
| | | A/G | 9 (9.4%) | 17 (12.2%) | 1.24 (0.46–3.32) | |
| **rs5185** | --- | T/T | 95 (99%) | 138 (99.3%) | 1.00 | 0.83 |
| | | T/G | 1 (1.1%) | 1 (0.7%) | 0.71 (0.03–17.17) | |
| **rs5186** | Log-additive | | | | 0.73 (0.47–1.15) | 0.17 |

Genotype- and allele type-specific risks obtained in the best model of inheritance based in Akaike information criterion (AIC). OR, odds ratio; CI, confidence interval; ICU, intensive care unit; SNPs, single nucleotide polymorphisms.

The A/A genotype for rs5183 showed a higher hospitalization risk in patients with comorbidities than without comorbidities patients (p<0.0001). The study of the interactions between the rest of SNPs in ACE2 or AGTR1 genes did not show statistical differences.

## The association of haplotype in the genes with hospitalization risk

None of the haplotypes of the four ACE2 tagSNPs or three AGTR1 tagSNPs were associated with the risk of hospitalization (p>0.05) or the severity of disease.

## Discussion

This study shows that comorbidities, older age and male gender were correlated with poor outcome in COVID-19 disease, as it has been previously reported [27]. Recently, Kouhpayeh HR et al. (2021) showed a significantly higher age and prevalence of diabetes and hypertension in COVID-19 patients with severe disease than a non-severe disease [28].

Different groups of severity were defined for comparisons, from outpatients, hospitalization, ICU admissions to death. Analysis of polymorphisms in the ACE2 and AGTR1 genes provided interesting associations with disease severity with potential for clinical stratification. Our study aimed to investigate if common genetic variants in key RAS genes impact clinical outcomes in COVID-19 infection.

Our study demonstrated that some SNPs in ACE2 were associated with COVID-19 disease. The genotype frequencies of rs2074192 and rs1978124 SNPs seen for heterozygosity in woman suggest a protective effect. It is noteworthy that the ACE2 is located on the X chromosome, causing the impossibility of heterozygosity in men. Therefore SNPs in their single copy could be related to the worst outcomes observed in males [29].

In addition, the association of the different ACE2 SNPs with the severity of the disease increased respect to the risk of hospitalization (**Table 3**) when the group of outpatients and the group of greater severity (ICU + deceased) were compared (**Table 4**).

ACE2 is involved in the balance of a system in which malfunctioning has been linked to a number of conditions including hypertension, myocardial infarction, heart failure, acute lung injury and diabetes mellitus [30]. In humans, several studies have shown a strong association of ACE2 polymorphisms with hypertension in female Chinese patients with metabolic syndrome [30] or essential hypertension [31, 32]. Thus, together with the biochemical data that has identified ACE2 as a negative regulator component of the RAS (ie, degrading Angiotensin II to generate Angiotensin 1–7), ACE2 can be thought to play a profound role in controlling blood pressure. Which suggests that those hypertensive women or with comorbidities related to COVID-19 are more susceptible to the changes produced by certain SNPs in the components of the RAS such as ACE2.

The level and expression pattern of ACE2 in different tissues and cells due to age, disease, or pharmacological therapy can be critical to the susceptibility and symptoms resulting from SARS-CoV-2 infection [14]. Zhou and collaborators [33] showed in heart failure disease an increased ACE2 expression, in which viral infection was related to a higher risk of myocardial infarction and worse outcome. On the other hand, a low expression of ACE2 leads to increased production of Angiotensin II, which can facilitate lung disease [34]. ACE2 genetic variations could be crucial to the susceptibility and course of COVID-19.

Recently, controversial results have been published on the role of different ACE2 SNPs in COVID-19 disease. Karakaş et al. (2021) failed to show any association between rs2106809 and rs2285666 and the clinical course of COVID-19 in a cohort of 155 patients [35]. On the other hand, Srivastava et al. (2020) published a positive correlation between rs2285666 and a lower infection rate as well as case-fatality rate among Indian populations [19]. In addition, it has been reported allele A of rs2285666 affect splice site which leads to an increase in the level of ACE2 protein in serum [36]. In keeping with the later, our results are consistent with an association of this variant with severity of COVID-19 manifestations in women (OR = 12.61, 95% CI: 1.26–125.87, p = 0.0081).

AGTR1 encodes the angiotensin type 1 receptor and is located in chromosome 3q24. Angiotensine II is a vasopressor hormone that regulates hypertrophy/hyperplasia, vascular cell migration, and the expression of pro-inflammatory genes. It acts mainly through AGTR1 to promote vascular muscle constriction. It is an important regulator of blood pressure and homeostasis in the cardiovascular system. Elevated tissue levels of Angiotensine II have been described in various pathological conditions, suggesting an important role in the pathogenesis of hypertension, cardiovascular diseases (myocardial infarction and arteriosclerosis) and kidney disease [37, 38]. Our results did not show significant differences between genotypes frequencies in AGTR1 SNPs and no association was identified in the hospitalized group regarding severity of the disease. Nevertheless, significant differences in A/A genotype frequencies in AGTR1 rs5183 were found between the groups with and without comorbidities regarding hospitalization risk, this suggest that hypertension or diabetes in patients with an specific genotype could increase the severity of COVID-19 disease.

Angiotensin II, the main effector of RAS, was shown to promote vascular senescence onset and progression, leading to age-related vascular diseases [39]. The AGTR1 receptor mediates its detrimental effects. Two SNPs (rs422858 and rs275653) in the AGTR1 promoter associated to reduced protein level, were significantly associated with the ability to attain extreme old age, that suggest their role in aging and age-related diseases [40].

The ACE I/D polymorphism has been associated with higher serum ACE levels [41], obesity [42], hypertension [43], increased cardiovascular risk [44], and thrombophilia [45]; all clinical conditions related with more aggressive COVID-19 disease [45].

In our study I/D polymorphism in ACE failed HWE in hospitalized group, which might be due to the association with the disease. The Covid-19 patients with comorbidities and DD genotype in ACE had a higher risk to be hospitalized (OR = 2.97, 95% CI: 1.23–7.17; p<0.05). In addition, when we analyzed the four severity groups independently, it was remarkable the association of the D/D with the deceased group. According with this, Delanghe et al (2020) reported the COVID-19-associated mortality correlated with D-allele of ACE [46] and the I allele decreased the risk of COVID-19 infection in other cohort of 504 subjects [28]. D/I polymorphism in ACE was also associated with ACE2 protein levels in lung tissue, thereby potentially affecting infectivity by SARS-CoV-2 [47].

Despite subgroup analysis suggested an association between some polymorphisms in ACE and AGTR1 and severity, multivariable analysis did not revealed consistent findings. Further studies with larger cohorts would be needed to definitively state the role of ACE and AGTR1 polymorphisms in COVID-19 disease and outcomes.

## Conclusions

Heterozygosity of rs2074192 and rs1978124 SNPs in ACE2 is associated with the disease severity caused by SARS-CoV-2 being a protection factor in women. On the other hand, the C/C genotype of rs2106809 and the allele A of rs2285666 in ACE2 are risk factors in patients with COVID-19. The different SNPs of ACE2 and rs5183 AGTR1 showed an association with severity and death in patients with COVID-19 and comorbidities.

## Supporting information

**S1 Table. Exact test for Hardy–Weinberg equilibrium (p-value).**
(DOCX)

**S2 Table. Genotype and allele frequencies of ACE, ACE2 and AGTR1 SNPs in outpatients and hospitalized Covid-19 cases, and genotype- and allele type-specific risks.**
(DOCX)

**S3 Table. Genotype and allele frequencies of ACE, ACE2 and AGTR1 SNPs in outpatients and ICU+deceased Covid-19 cases, and genotype- and allele type-specific risks.**
(DOCX)

**S4 Table. Relationship of comorbidities with the different SNPs in COVID-19 patients.**
Interaction analysis with comorbidities and different SNPs.
(DOCX)

**S5 Table. Relationship of comorbidities with the different SNPs in outpatients and ICU+-deceased COVID-19 patients.** Interaction analysis with comorbidities and different SNPs.
(DOCX)

## Acknowledgments

We do thank the collaboration of Doctors Encarnación Moral, Alicia Hernández Torres, Helena Albendín, Aylen Roura and Carlos Galera from the Infectious Disease Unit; Silvia Sánchez, Carlos Albacete, José Higino de Gea from Intensive Care Unit; Joaquin Palomar from the Consejería de Salud and Juan Antonio Gómez Company from the Servicio Murciano de Salud for their support in the identification and the collection of cases. We do appreciate the generosity of the patients included in the study and their relatives.

The authors wish to thank the participating centers' biobanks (BIOBANC-Mur/Instituto Murciano de Investigación Biosanitaria, PT20/00109; Biobank Hospital Universitario La Fe, PT17/0015/0043; Biobank Hospital UniversitarioPuertade Hierro Majadahonda (HUPHM)/ Instituto de Investigación Sanitaria Puerta de Hierro-Segovia de Arana (IDIPHISA), PT17/ 0015/0020 in the Spanish National Biobanks Network) for the human specimens used in this study.

## Author Contributions

**Conceptualization:** Elisa García Vázquez, Fernando Domínguez Rodriguez.

**Data curation:** Elisa Nicolás Rocamora, Asunción Iborra Bendicho, Elisa García Vázquez, Esther Zorio, Fernando Domínguez Rodriguez, Cristina Gil Ortuño, Ana Isabel Rodríguez, Antonio J. Sánchez-López, Rubén Jara Rubio, Antonio Moreno-Docón, Pedro J. Marcos, Pablo García Pavía, Roberto Barriales Villa.

**Formal analysis:** Maria Sabater Molina, Cristina Gil Ortuño, Juan R. Gimeno Blanes.

**Funding acquisition:** Elisa García Vázquez, Juan R. Gimeno Blanes.

**Investigation:** Maria Sabater Molina, Cristina Gil Ortuño, Pablo García Pavía, Juan R. Gimeno Blanes.

**Methodology:** Maria Sabater Molina, Ana Isabel Rodríguez, Juan R. Gimeno Blanes.

**Project administration:** Maria Sabater Molina, Juan R. Gimeno Blanes.

**Resources:** Asunción Iborra Bendicho, Fernando Domínguez Rodriguez, Antonio J. Sánchez-López, Rubén Jara Rubio, Antonio Moreno-Docón, Pedro J. Marcos, Pablo García Pavía, Roberto Barriales Villa.

**Validation:** Pedro J. Marcos, Roberto Barriales Villa, Juan R. Gimeno Blanes.

**Writing – original draft:** Maria Sabater Molina, Elisa García Vázquez, Esther Zorio, Fernando Domínguez Rodriguez, Pablo García Pavía, Juan R. Gimeno Blanes.

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
