## [Decision Letter · Decision Letter 0]

19 Nov 2021

PONE-D-21-32344Polymorphisms in ACE, ACE2, AGTR1 genes and severity of COVID-19 disease.PLOS ONE

Dear Dr. Sabater-Molina,

Thank you for submitting your manuscript to PLOS ONE. After careful consideration, we feel that it has merit but does not fully meet PLOS ONE’s publication criteria as it currently stands. Therefore, we invite you to submit a revised version of the manuscript that addresses the points raised during the review process.

Please pay particular attention to the  Reviewer1' s comments:  check that all protocols are correctly reported and explain the discrepancies in the number of analysed subjects.

We look forward to receiving your revised manuscript.

Kind regards,

Cinzia Ciccacci

Academic Editor

PLOS ONE

Journal Requirements:

3. We note that you have included the phrase “data not showed” in your manuscript. Unfortunately, this does not meet our data sharing requirements. PLOS does not permit references to inaccessible data. We require that authors provide all relevant data within the paper, Supporting Information files, or in an acceptable, public repository. Please add a citation to support this phrase or upload the data that corresponds with these findings to a stable repository (such as Figshare or Dryad) and provide and URLs, DOIs, or accession numbers that may be used to access these data. Or, if the data are not a core part of the research being presented in your study, we ask that you remove the phrase that refers to these data.

Reviewers' comments:

Reviewer's Responses to Questions

**Comments to the Author**

1. Is the manuscript technically sound, and do the data support the conclusions?

Reviewer #1: Yes

Reviewer #2: Yes

2. Has the statistical analysis been performed appropriately and rigorously? 

Reviewer #1: Yes

Reviewer #2: Yes

3. Have the authors made all data underlying the findings in their manuscript fully available?

Reviewer #1: Yes

Reviewer #2: Yes

4. Is the manuscript presented in an intelligible fashion and written in standard English?

Reviewer #1: Yes

Reviewer #2: Yes

5. Review Comments to the Author

Reviewer #1: In the manuscript, the authors present sound evidence regarding ACE23 polymorphisms and COVID-19. The statistical power of trhe study is sufficient and the agreement with the Hardy-Weinberg equilibrium has been tested.

In the discussion , the existing literature between ACE1 D/I polymorphism and COVID-19 and the relationship between ACE1 polymorphism and ACE2 in COVID-19 should be included ( refs: Delanghe JR, Speeckaert MM, Marc L De Buyzere ML. COVID-19 infections are also affected by human ACE1 D/I polymorphism. Clin Chem Lab Med 2020;58: 1125-1126. and

Jacobs M, Lahousse L, Van Eeckhoutte HP, Wijnant SRA, Delanghe JR, Brusselle GG, Bracke KR. Effect of ACE1 polymorphism rs1799752 on protein levels of ACE2, the SARS-CoV-2 entry receptor, in alveolar lung epithelium. ERJ Open Res 2021; 7: 00940-2020. )

Reviewer #2: The Authors investigated the associations between 8 SNPs from ACE, ACE2 and AGTR1 genes and disease severity, in a population of COVID-19 subjects categorised as outpatients, hospitalised on wards, admitted to the ICU and deceased. Comorbidity data were considered for proper adjustment. Phenotype distribution of selected SNPs were analysed and correlated with disease severity. This is an interesting manuscript although some changes should be addressed

Abstract:

Results should report also the data about ACE and AGTR1 analysis.

line 44: 318 patients with 104 outpatients were classified.

Introduction /aim:

lines 85-87: Please add the reference for data on symptomatic and asymptomatic subjects showing the variants specific for ACE, ACE2 and AGTR1 genes, or others.

lines 96-97: please specify all SNPs investigated.

methodology:

This part is missing in several information regarding the protocols and kits used for the study.

line 108: 317 COVID-19 subjects: I'm confusing about the number of total COVID-19 subjects included in the study, please provide right number.

line 108: specify the kit used for PCR test

line 115: specify the volume of blood collected and the test used for DNA extraction.

line 116: specify PCR and electrophoresis conditions

results:

line 157: again, how many patients did author include in this study? 317 or 318.

discussion: this section needs a more focused discussion

line 312 some references need typing corrections (see lines 334 and 418).

finally provide an accurate revision of literature regarding these three genes.

6. PLOS authors have the option to publish the peer review history of their article (what does this mean?). If published, this will include your full peer review and any attached files.

Reviewer #1: No

Reviewer #2: No

---

## [Author Response · Author response to Decision Letter 0]

15 Dec 2021

Response to comments of editor:

Comments to the Author

[1] Journal Requirements:

3. We note that you have included the phrase “data not showed” in your manuscript. Unfortunately, this does not meet our data sharing requirements. PLOS does not permit references to inaccessible data. We require that authors provide all relevant data within the paper, Supporting Information files, or in an acceptable, public repository. Please add a citation to support this phrase or upload the data that corresponds with these findings to a stable repository (such as Figshare or Dryad) and provide and URLs, DOIs, or accession numbers that may be used to access these data. Or, if the data are not a core part of the research being presented in your study, we ask that you remove the phrase that refers to these data.

[1] We do appreciate editor and reviewers´ positive comments on our paper. We have reviewed the manuscript and address the style requirements. Comprehensive information related to the investigation has been provided in figures and tables (including 5 supplementary tables). We have accordingly removed the line “data not showed” in the new version of the manuscript. 

Response to comments of reviewer #1:

We thank reviewer 1 for the review and comments.

Comments to the Author

[2] Reviewer #1: In the manuscript, the authors present sound evidence regarding ACE2 3 polymorphisms and COVID-19. The statistical power of trhe study is sufficient and the agreement with the Hardy-Weinberg equilibrium has been tested.

In the discussion , the existing literature between ACE1 D/I polymorphism and COVID-19 and the relationship between ACE1 polymorphism and ACE2 in COVID-19 should be included ( refs: Delanghe JR, Speeckaert MM, Marc L De Buyzere ML. COVID-19 infections are also affected by human ACE1 D/I polymorphism. Clin Chem Lab Med 2020;58: 1125-1126. and

Jacobs M, Lahousse L, Van Eeckhoutte HP, Wijnant SRA, Delanghe JR, Brusselle GG, Bracke KR. Effect of ACE1 polymorphism rs1799752 on protein levels of ACE2, the SARS-CoV-2 entry receptor, in alveolar lung epithelium. ERJ Open Res 2021; 7: 00940-2020. )

[2] We thank the reviewer #1. We have included a new paragraph in the discussion section (page 19, lines 355-359) of the manuscript and these references have been added as new valuable references 47 and 48 following the recommendations. 

Response to comments of reviewer #2:

We thank reviewer #2 for the review and very appropriate comments.

Comments to the Author

Reviewer #2: The Authors investigated the associations between 8 SNPs from ACE, ACE2 and AGTR1 genes and disease severity, in a population of COVID-19 subjects categorised as outpatients, hospitalised on wards, admitted to the ICU and deceased. Comorbidity data were considered for proper adjustment. Phenotype distribution of selected SNPs were analysed and correlated with disease severity. This is an interesting manuscript although some changes should be addressed

Abstract:

[3] Results should report also the data about ACE and AGTR1 analysis.

[3] Comprehensive information related to the investigation has been provided in figure and tables (including 5 supplementary tables). We have accordingly included additional information on the results of ACE and AGTR1 analysis in the in the revised version of the manuscript (page 4, lines 66-70, and page 19, lines 356-359).

[4] line 44: 318 patients with 104 outpatients were classified. 

[4] We apologize, but we can not fully understand this point. As states in the methods section (page 6, section Study subjects) 

“A total of 318 COVID-19 subjects with positive polymerase chain reaction (PCR) test for SARS-Cov-2 virus were included in the study(…). The participants were grouped into 4 groups: outpatients cured, hospitalized on the wards, admitted to the Intensive Care Unit (ICU) and deceased as a result of the infection or its complications. Patients were selected consecutively from those with available samples from the 5 participating centers’ biobanks, with the aim to achieve a minimum of 50 cases per group.”

All patients were evaluated and tested PCR SARS-Cov-2 positive in either (1) at outpatient’s clinics (at the local surgeries by their General Practitioner) or (2) at the hospital admission. Outpatients after the diagnosis of COVID-19 was achieved and considered low risk, then, they were followed-up by their GPs with close monitoring (daily phone calls and home/ surgery visits where required). 

We believe line 44 of the abstract was correct. We have only changed “classified” for “grouped” and “;” for a “,” after “outpatients”, for clarity (new lines 50 and 51).

This paragraph now reads: 

“318 (aged 59.6±17.3 years, males 62.6%) COVID-19 patients were grouped based on the severity of symptoms: Outpatients (n = 104, 32.7%), hospitalized on the wards (n = 73, 23.0%), Intensive Care Unit (ICU) (n = 84, 26.4%) and deceased (n = 57, 17.9%).”

Introduction/aim:

[5]lines 85-87: Please add the reference for data on symptomatic and asymptomatic subjects showing the variants specific for ACE, ACE2 and AGTR1 genes, or others.

[5] Thank you for the comments. The reference has been added as new reference 10 (page 5, line 96).

10. Anastassopoulou C, Gkizarioti Z, Patrinos GP, Tsakris A. Human genetic factors associated with susceptibility to SARS-CoV-2 infection and COVID-19 disease severity. Hum Genomics. 2020 Oct 22;14(1):40. doi: 10.1186/s40246-020-00290-4. PMID: 33092637; PMCID: PMC7578581.

[6]lines 96-97: please specify all SNPs investigated.

[6] The different eight SNPs have been specified accordingly in the new version of the manuscript (page 5, last paragraph, lines 102-103).

methodology:

This part is missing in several information regarding the protocols and kits used for the study. 

[7] line 108: 317 COVID-19 subjects: I'm confusing about the number of total COVID-19 subjects included in the study, please provide right number.

[7] Thank you very much for the comment. The total COVID-19 subjects included in the study were 318. Any discrepancy in the manuscript has been reviewed and corrected in the text and in the tables.

line 108: specify the kit used for PCR test[8]

[8] The kit used for PCR test was Novel Coronavirus (2019-nCoV) Real Time Multiplex RT-PCR kit (Detection for 3 Genes), manufactured by Shanghai ZJ Bio-Tech Co., Ltd. (Liferiver) and CFX96 Touch Real-Time PCR Detection System (BioRad) (page 6, last paragraph, lines 119-121).

line 115: specify the volume of blood collected and the test used for DNA extraction.[9]

[9] The volume of blood and the kit of DNA extraction have been specified (page 7, first paragraph, lines 127-128): DNA was extracted from 400 µl of peripheral blood using the Maxwell® 16 Blood DNA Purification Kit (Promega). 

line 116: specify PCR and electrophoresis conditions[10]

[10] The PCR conditions and percentage of agarose used to electrophoresis have been specified (page 7, lines 129-140).

results:

[11]line 157: again, how many patients did author include in this study? 317 or 318. 

[11] The total COVID-19 subjects included in the study were 318. Any discrepancy in the manuscript has been corrected.

line 312 some references need typing corrections (see lines 334 and 418). [12]

[12] All references have been reviewed and corrected following reviewer’s comments.

[13] discussion: this section needs a more focused discussion

finally provide an accurate revision of literature regarding these three genes.

[13] Following reviewer’s comments, this section has been amended and new references have been added to support discussion. An updated revision of literature has been performed with the inclusion of 7 new references.

---

## [Decision Letter · Decision Letter 1]

13 Jan 2022

Polymorphisms in ACE, ACE2, AGTR1 genes and severity of COVID-19 disease.

PONE-D-21-32344R1

Dear Dr. Sabater-Molina,

We’re pleased to inform you that your manuscript has been judged scientifically suitable for publication and will be formally accepted for publication once it meets all outstanding technical requirements.

Kind regards,

Cinzia Ciccacci

Academic Editor

PLOS ONE

Additional Editor Comments (optional):

Reviewers' comments:

Reviewer's Responses to Questions

**Comments to the Author**

1. If the authors have adequately addressed your comments raised in a previous round of review and you feel that this manuscript is now acceptable for publication, you may indicate that here to bypass the “Comments to the Author” section, enter your conflict of interest statement in the “Confidential to Editor” section, and submit your "Accept" recommendation.

Reviewer #1: All comments have been addressed

Reviewer #2: (No Response)

2. Is the manuscript technically sound, and do the data support the conclusions?

Reviewer #1: Yes

Reviewer #2: Yes

3. Has the statistical analysis been performed appropriately and rigorously? 

Reviewer #1: Yes

Reviewer #2: Yes

4. Have the authors made all data underlying the findings in their manuscript fully available?

Reviewer #1: Yes

Reviewer #2: Yes

5. Is the manuscript presented in an intelligible fashion and written in standard English?

Reviewer #1: Yes

Reviewer #2: Yes

6. Review Comments to the Author

Reviewer #1: the authors of the revised manuscript have replied in an adequate manner to the questions and the comments raised by the reviewer

Reviewer #2: The manuscript has been significantly improved in both methodology and data presentation, and the importance of the investigated polymorphisms appears clearly in the discussion section.

7. PLOS authors have the option to publish the peer review history of their article (what does this mean?). If published, this will include your full peer review and any attached files.

Reviewer #1: **Yes: **Joris Delanghe

Reviewer #2: No

---

## [Editor Report · Acceptance letter]

18 Jan 2022

PONE-D-21-32344R1 

Polymorphisms in ACE, ACE2, AGTR1 genes and severity of COVID-19 disease. 

Dear Dr. Sabater-Molina:

I'm pleased to inform you that your manuscript has been deemed suitable for publication in PLOS ONE. Congratulations! Your manuscript is now with our production department. 

Kind regards, 

on behalf of

Dr. Cinzia Ciccacci 

Academic Editor

PLOS ONE